# Exploring Mediation Effects of Gait Speed, Body Balance, and Falls in the Relationship between Physical Activity and Health-Related Quality of Life in Vulnerable Older Adults

**DOI:** 10.3390/ijerph192114135

**Published:** 2022-10-29

**Authors:** Marcelo de Maio Nascimento, Élvio Rúbio Gouveia, Bruna R. Gouveia, Adilson Marques, Cíntia França, Duarte L. Freitas, Pedro Campos, Andreas Ihle

**Affiliations:** 1Department of Physical Education, Federal University of Vale do São Francisco, Petrolina 56304-917, Brazil; 2Department of Physical Education and Sport, University of Madeira, 9020-105 Funchal, Portugal; 3LARSYS, Interactive Technologies Institute, 9020-105 Funchal, Portugal; 4Center for the Interdisciplinary Study of Gerontology and Vulnerability, University of Geneva, 1205 Geneva, Switzerland; 5Regional Directorate of Health, Secretary of Health of the Autonomous Region of Madeira, 9004-515 Funchal, Portugal; 6Saint Joseph of Cluny Higher School of Nursing, 9050-535 Funchal, Portugal; 7CIPER, Faculty of Human Kinetics, University of Lisbon, 1495-751 Lisbon, Portugal; 8ISAMB, Faculty of Medicine, University of Lisbon, 1649-020 Lisbon, Portugal; 9Centre of Research, Education, Innovation and Intervention in Sport (CIFI2D), Faculty of Sport, University of Porto, 4200-450 Porto, Portugal; 10Department of Informatics Engineering and Interactive Media Design, University of Madeira, 9020-105 Funchal, Portugal; 11Department of Psychology, University of Geneva, 1205 Geneva, Switzerland; 12Swiss National Centre of Competence in Research LIVES—Overcoming Vulnerability: Life Course Perspectives, 1015 Lausanne, Switzerland

**Keywords:** aging, quality of life, physical activity, gait speed, body balance, falls, vulnerability

## Abstract

The present study aimed to examine whether gait speed (GS), body balance (BB), and falls mediated the relationship between physical activity (PA) and health-related quality of life (HRQoL) in community-dwelling older adults. This is a cross-sectional study that included 305 men and 314 women (69.5 ± 5.6 years), residing in the Autonomous Region of Madeira, Portugal. HRQoL and PA were assessed using the SF-36 and Baecke Questionnaires, respectively. While BB was obtained by the Fullerton Advance Balance (FAB) scale, GS by the 50-foot (15 m) walk test, and the frequency of falls was obtained by self-report. According to the analyses, when GS and BB were placed concomitantly as mediators, the direct effect revealed by the model revealed a non-significant relationship between PA and falls. Thus, in the context of falls, GS and BB partially mediated the association between PA and HRQoL in approximately 29.7%, 56%, and 49.2%, respectively. The total HRQoL model explained a variance of 36.4%. The results can help to understand the role that GS, BB, and falls play in the relationship between PA and HRQoL of the vulnerable older adult population.

## 1. Introduction

Physical activity (PA) and health-related quality of life (HRQoL) are two crucial components in determining the health status of an individual, especially during the aging process [1,2]. PA is defined as a set of activities performed by individuals in their daily household activities and participation in sports and leisure activities [3]. According to the latest guidelines on physical activity and sedentary behavior, issued by the World Health Organization [4], to benefit health substantially, older adults should do at least 150 min of moderate-intensity aerobic activity or 75 min of vigorous-intensity aerobic activity, or have an equivalent combination of both. In addition, perform muscle strengthening and balance activities for three or more days to prevent falls. Based on PA levels, it is possible to estimate daily energy expenditure and, consequently, the level of the sedentary lifestyle of an individual [5]. Therefore, a high level of PA reflects a more active lifestyle, which, in turn, increases the probability that the individual will both maintain and improve their physical capacity [6]. Thus, in the case of the elderly population, having an adequate level of PA will reflect positively on their health status, prolonging their autonomy [7,8]. From this perspective, health risks increase when PA levels are low, which reflects a sedentary lifestyle, in general, associated with low caloric expenditure. Consequently, musculoskeletal neuromotor and cardiorespiratory problems may arise [7]. In turn, a reduction in PA levels can negatively affect HRQoL [1,9].

HRQoL is a multidimensional term that integrates a series of perceptions of the individual’s physical, mental, and social symptoms, including the limitations caused by diseases [10]. Thus, depending on the perception formed by the individual about their general state of health, this may have a positive or negative influence on their physical and/or emotional well-being [11]. The literature highlights that with advancing age, HRQoL tends to decrease [12]. This fact highlights the importance of creating measures capable of promoting the levels of physical and mental health of older adults. Previous studies have highlighted that PA and HRQoL, in addition to being associated, are not fundamental for the individual to age in a healthy way, which contributes to a lower risk of disease [13,14]. This includes preventing mobility limitations, which increase the risk of disability in many older adults [15]. Therefore, adequate levels of PA are important for the older adult population to maintain their physical functions, which include gait speed (GS) and body balance (BB) [3,16]. GS and BB are fundamental motor skills for safely performing activities of daily living (ADLs) [17]. A slow gait, associated with the BB deficit, is a contributing factor to the vulnerability of older adults and increased risk of falls [18,19].

When it comes to the older adult population, falls represent a public health problem [20], due to their association with injuries, pain, and hospitalization days, which are responsible for high costs [21]. After a fall, there is an increased chance of an older adult falling again [22]. Moreover, falls are responsible for fractures, which can limit ADL performance [23]. Therefore, depending on the severity of the fall, older adults are at risk of death [24]. In this context, during vulnerable aging, falls negatively impact the perception of HRQoL [25,26]. Over the years, studies have reported the benefits for older adults of having high levels of PA for both GS performance [27,28] and BB control [29,30], and consequently improving HRQoL [1,31]. In this perspective, it is important to highlight that in a current previous investigation [30], the authors of the present study performed a mediation analysis to examine whether lower limb muscle strength (LEMS) and BB mediated the relationship between PA and HRQoL. The results showed that the mediating role of LEMS and BB in the association was approximately 39.6% and 47%, respectively. These outcomes brought to light new research questions. 

Thus, knowing that there is a direct relationship between low levels of PA and the performance deficit of GS and BB, which in turn, increases the rate of an older adult falling [32,33], and reflects negatively on HRQoL [34,35], this study aimed to examine whether and how in detail GS, BB and falls mediate the relationship between PA and HRQoL in community-dwelling older adults. With regard to the novel contribution of this study, to the best of our knowledge, to date, no study has examined the proportion of mediation of GS, BB, and declines in the relationship between PA and HRQoL of older adults. We hypothesize that there is (1) a positive relationship between PA with GS and PA and BB and, as a result, a negative relationship between GS and BB with falls, (2) falls have a negative relationship with HRQoL, and (3) PA through GS and BB will show a positive relationship on HRQoL.

## 2. Methods

### 2.1. Sample and Study Design

This is a cross-sectional analytical observational investigation, which included 619 individuals (69.5 ± 5.6 years). Of these, 305 were men and 314 were women, residing in the Autonomous Region of Madeira, Funchal, Portugal. Participants were recruited in clubs, cultural and sports associations, older adult homes, street markets, and churches, as well as through publicity in local newspapers, radio, and television. Inclusion criteria were as follows: (1) age 60–89 years, (2) domicile in the local community, and (3) ability to walk independently. As exclusion criteria, we adopted (1) impediment to participating in the set of assessments and (2) medical contraindications to performing physical exercises with submaximal intensity [36]. The procedures were approved by the Scientific and Ethics Committee and the Department of Physical Education and Sports of the University of Madeira (UMa), in addition to receiving authorization from the Regional Secretariat of the Social Affairs Committees and had the support of the FCT (SFRH/BD29300/2006). Before carrying out the assessments, the participants were informed about the procedures and signed a consent form. Study procedures followed the Declaration of Helsinki. The evaluations were carried out in 2009 by researchers trained in the Human Physical Growth and Motor Development Laboratory at UMa.

### 2.2. Measures

#### 2.2.1. Demographics and Health Profile

Sociodemographic information (i.e., age, sex), comorbidities, and daily medication consumption were collected through face-to-face interviews.

#### 2.2.2. Falls

The participants’ history of falls was obtained through face-to-face interviews. The questions asked were the following: (1) Have you suffered any falls in the last 12 months? (binary answer: yes or no); and (2) If yes, could you tell us the total number of falls in the last 12 months? In the present study, we assumed the first question for the formation of the faller and non-faller groups.

#### 2.2.3. Anthropometry

Height and body mass were measured using a Welmy^®^ anthropometric scale (London, UK) with a stadiometer attached to 0.1 cm and 0.1 kg [37]. Afterward, the body mass index (BMI) was defined through the equation: weight (kg)/[height (m)]^2^.

#### 2.2.4. Physical Activity

Physical activity was estimated through the modified version of the Baecke Questionnaire for older adults [38]. The instrument is used worldwide, presenting three sections that assess the rates of physical activity in the last 12 months: (1) household activities (PA-domestic work); (2) leisure time activities (PA-leisure); (3) sporting activities (PA-sport), including regular weekly activities performed for at least one hour. For the evaluations, the total score (PA-total) was used, composed of the sum of the scores between PA-housework, PA-sport and PA-leisure. In their validation study, a time interval of three months, performed with 90 healthy, independent individuals (age 63 to 80 years), analyzes showed correlation reliability indices of 0.88, 0.81, and 0.74 for PA-work, PA-sport, and PA-leisure, respectively. The relative validity verified between the questionnaire and two reference methods, physical activity recall and pedometer, indicated a Spearman correlation coefficient of 0.78 and 0.72, respectively [38].

#### 2.2.5. Gait

GS was assessed using the 50 ft (15 m) walk test proposed by the Senior Fitness Test (SFT) battery [39]. Initially, participants were asked to walk at their preferred speed and performed the test three times. Finally, the best performance was computed for analysis. This test showed high test–retest reliability for households (0.94), and women (0.88), as well as a coefficient of validity for men (0.79), and for women (0.80) [39].

#### 2.2.6. Body Balance

The Fullerton Advanced Balance (FAB) battery was used to evaluate the performance of the BB [40]. Its protocol includes ten tasks, which allow for evaluating the performance of the older adult’s dynamic and static balance. The tasks consist of: (1) standing with feet together, and eyes closed, (2) reaching out to pick up an object (pencil) at shoulder height with the arm extended, (3) performing left and right 360° turn, (4) stepping up and down a six-inch bench, (5) walking in tandem, (6) standing on one leg, (7) with eyes closed, standing on foam, (8) perform a long jump, (9) walk and turn head to the side, and (10) recover from an unexpected loss of balance. Each task was scored on an ordinal scale of 4 points (0–4), with a maximum score of 40 points. Instructions on its application and equipment, including instructional video, were reported by Rose [39]. The FAB scale showed high test–retest reliability (0.96), as well as intra-examiner (0.92–1.00) and inter-examiner (0.91–0.95) reliability.

#### 2.2.7. Health-Related Quality of Life

HRQoL was assessed using the 36-item Short-Form Health Survey (SF-36) [41]. This questionnaire has eight dimensions, distributed in two components, each with four domains: (1) Physical: physical functioning (PF), physical role (RP), bodily pain (BP), general health status (GH), and (2) Mental: vitality (VT), social functioning (SF), emotional role (ER), and mental health (MH). The scores for each specific domain range from 0 to 100. Thus, the higher the score, the better the HRQoL. In the present study, we adopted a global and continuous measure of HRQoL, obtained by adding the scores of the physical and mental components of the SF-36 questionnaire. The instrument presented validity in the range of 0.92–0.95, and acceptable reliability for the eight dimensions with Cronbach’s alpha ranged from 0.62 to 0.94, in addition to test–retest coefficients between 0.43 and 0.90.

#### 2.2.8. Covariates

The following variables were considered confounding factors: gender, age, number of medications consumed per day, vision, BMI, hypertension, and musculoarticular problems. Additionally, therefore, controlled in the serial analysis of mediation.

### 2.3. Statistical Analysis 

Data distribution was analyzed using the Kolmogorov–Smirnov test. Considering that the data presented a normal distribution, continuous variables (i.e., age, medications, weight, height, number of falls over 12 months, BMI, GS, BB, and HRQoL) were presented as mean and standard deviation (SD). Categorical variables (i.e., sex, falls, and comorbidities) were presented by frequencies and percentages. Comparisons of significance levels between continuous variables were established using the unpaired Student’s *t*-test for independent samples, while statistical differences between categorical variables were tested by the chi-square test with Fischer’s correction. Although our study did not focus on drawing comparisons between groups, the main characteristics of the participants were presented in two groups for descriptive purposes: fallers and non-fallers (dichotomous variable). On the other hand, for serial analysis of parallel mediation (investigation goal), the mediator variable falls (*M*_3_) was formed by the total number of falls reported by participants over the last 12 months (scalar variable).

To achieve the study’s main objective, we conducted Parallel and Serial Mediation [42]. Before the mediation analyses, the assumptions of multiple regression were verified: the linear relationship between independent and dependent variables, mean of residuals equal to zero, normality of residuals, non-multicollinearity, non-autocorrelation of residuals, and homoscedasticity of residuals, or an equal variation [43]. Figure 1 illustrates the path model used, composed of 9 direct effects (paths a–i). The model included two parallel mediators (*M*_1_ and *M*_2_), preceded by a third mediator (*M*_3_). This type of analysis broadens and qualifies the understanding of how an independent variable (PA) can directly influence a dependent variable (HRQoL) mediated in parallel series by two mediators (GS and BB), which in turn indirectly affect falls (*M*_3_), and concluding the cycle, falls also influences the perception of HRQoL. Our analysis considered a complete mediation, if with the inclusion of GS and BB objectively measured (mediator variable), preceded serially by a third mediator (Falls), the size of the association between the independent variable (PA) and the dependent variable (HRQoL) did not remain significant, which would be indicated when the confidence interval included the value of zero [42]. On the other hand, a partial mediation would occur if the observed relationship between the independent variable (PA) and the dependent variable (HRQoL) became weaker after the inclusion of the mediating variables (*M*_1_, *M*_2_, *M*_3_).

The effects represented by the regression coefficients were estimated using PROCESS v4.0, a computational extension of the SPSS program (IBM, Chicago, IL, USA). For parallel multiple serial mediation analysis, we used Model 80 [42]. The coefficients (a–i) described in the equation (Figure 2) were calculated using least squares regression analyses. The direct and indirect effects of *X* on *Y* of the model were estimated using four equations, one for each of the three mediators, and one for the final consequent *Y*, as follows:*M*_1_ = i *M*_1_ + a1*X* + e *M*_1_

*M*_2_ = i *M*_2_ + a2*X* + d21 *M*_1_ + e *M*_2_

*M*_3_ = i *M*_3_ + a3*X* + d31 *M*_1_ + e *M*_3_

*Y* = i*Y* + c′*X* + b1*M*_1_ + b2*M*_2_ + b3*M*_3_ + e*Y*


The analysis procedures of the complete model (see Figure 2 for better visualization) were composed of four submodels, namely: Models 1 and 2, which perform the regression of each of the parallel mediators (*M*_1_ and *M*_2_) in *X*; Model 3 processed the regression of *M*_3_ simultaneously on *X*, *M*_1_ and *M*_2_; and Model 4 performed the regression from *Y* to *X*, *M*_1_, *M*_2_, and *M*_3_. In parallel, the complete model calculated specific indirect effects (SIE’s). The principle used in this calculation was the product of the path coefficients in a sequence. The SIE’s calculated between *X* and *Y* were: SIE1 = a*b, SIE2 = a*g*d, SIE3 = c*d, SIE4 = e*f, SIE5= e*h*d. Finally, the total indirect effect corresponded to the sum of all SIE’s = SIE1 + SIE2 + SIE3 + SIE4 + SIE5. Our mediation hypothesis was estimated using a confidence interval (95% CI) with bias correction and acceleration (BCa) by the Bootstrapping method; we set the number of bootstrap samples at 5000. When the confidence interval of the equation did not include zero, specific indirect effects were assumed to be significant. [44]. The following procedure was used to calculate the proportion of mediation effects: subtraction of 1 minus the result of the division between the direct effect and the total effect [42]. The results illustrated in the figures correspond to the standardized parameters *β*. The significance level was set at 5%.

## 3. Results

### 3.1. Participants’ Characteristics

Among the 619 participants (Table 1), 314 were women (69.49 ± 5.57 years old) and 305 men (69.51 ± 5.69 years old) (*p* < 0.001). The observed frequency of falls was 1–10 episodes.

### 3.2. Serial and Parallel Mediation Analysis

The model was significant (Figure 2). The analysis of the three mediation variables proved them to be predictors of the relationship between PA and HRQoL: F(1.614) = 101.1217, *p* < 0.001, R^2^ = 0.32. The results confirmed Hypothesis 1: (SIE1) PA affected positively and significantly GS (*β* = 0.08, *t* (614) = 10.056, *p* < 0.001), which in turn acted negatively and significantly on falls (*β* = −0.92, *t* (612) = −3.026, *p* = 0.026), and (SIE4) PA affected positively and significantly BB (*β* = 1.52, *t* (614) = 8.120, *p* < 0.001), which in turn acted negatively and significant on falls (*β* = −0.05, *t* (612) = −0.046, *p* < 0.001). Subsequently, the analysis confirmed Hypothesis 2: (SIE3) the parallel paths GS and BB mediated by falls affected negatively and significantly HRQoL (*β* = −2.24, *t* (611) = −5.484, *p* < 0.001). Finally, we confirmed Hypothesis 3: (SIE2) the path PA through GS affected positively and significantly HRQoL (*β* = 13.75, *t* (611) = 4.426, *p* < 0.001), and (SIE5) PA through BB also acted positively and significantly on HRQoL (*β* = 0.63, *t* (611) = 5.957, *p* < 0.001). 

Regarding the direct effect (path i = *X* − *Y*), there was a negative and significant relationship between PA and HRQoL (*β* = −1.84, *t* (611) = 3.245, *p* = 0.012). On the other hand, the total effect of the model (*X* − *Y*) revealed a positive and significant association between PA and HRQoL (*β* = 4.20, *t* (611) = 7.182, *p* < 0.001). Therefore, the analysis based on 5000 bootstrap samples indicated positive and significant results for the indirect effects of the following paths: PA-GS-HRQoL (*β* = 1.0550, 95% CI BCa = 0.5373–1.5888), PA-BB-HRQoL (*β* = 1.1680, 95% CI BCa = 0.6713–1.7290), PA-GS-Falls-HRQoL (*β* = 0.1590, 95% CI BCa = 0.0443–0.3062), and PA-BB-Falls-HRQoL (*β* = 0, 1920, 95% CI BCa = 0.0989–0.3310). A negative and non-significant association was indicated by the general indirect route of PA for HRQoL (*β* = −0.2167, 95% CI BCa = −0.4849–0.0202), noting that GS, BB and falls proved to be independent mediators of effect that PA exerts on the HRQoL of older adults

The proportion of the total effect that PA had on HRQoL mediated by GS and BB through falls was up to 36.4%. Regarding the mediation proportion, the variables GS, BB, and falls explained the variance of the association between PA and HRQoL at approximately 29.7%, 56%, and 49.2%, respectively.

## 4. Discussion

This study examined, in a large sample of community-dwelling older adults, whether GS, BB, and falls mediated the relationship between PA and HRQoL. A parallel–serial mediation analysis was performed, and the total HRQoL variance explained by the entire model was 36.4%. In this scope, our three hypotheses were confirmed. We found that GS and BB inversely affect falls and, in turn, falls inversely affect HRQoL. When GS and BB were placed as mediators simultaneously, controlling for covariates (i.e., sex, age, medication, vision, hypertension, musculoarticular problems, and BMI), the direct effect of the pathway between PA and HRQoL (*x*-*y*) became non-significant. Thus, it was found that GS, BB, and falls partially mediated the link between PA and HRQoL by up to 29.7%, 56%, and 49.2%, respectively. The results suggested, in percentage terms, the effect that high levels of PA have on motor skills such as GS and BB, which are crucial for the older adult population to reach and/or maintain appropriate levels of HRQoL. Moreover, when these levels are adequate, it is possible to reduce both the risk of falls and the negative impact they have on HRQoL. As far as we know, this study is the first to present results in proportional terms on the investigated topic.

After controlling for potential confounding factors (i.e., sex, age, medication, vision, hypertension, musculoarticular problems, and BMI), the serial mediation model showed a negative association between GS and BB with falls. Thus, reductions in GS of approximately 0.1 m/s in the standard deviation (SD) represented an increase in the risk of falling by up to 0.92% (*M*_1_–*M*_3_, path h). Bohannon [45] suggested normal GS performance values of 1.13 m/s and 1.26 m/s for healthy individuals aged 70 to 79 years (both sexes), respectively. Additionally, for those aged 80 to 89 years, the suggested values were 0.94 and 0.97 m/s, respectively. According to our analysis, the average performance of both groups for GS was 1.24 m/s, and the difference between them was relatively small (Δ = 0.08 m/s). The literature suggests the value of 1 m/s as a cutoff point in the GS exam to identify the risk of falling in older adults [46]. Thus, GS values below 1 m/s are considered idle speed. In turn, the relationship between *M*_1_–*M*_3_ (path g) indicated that for each reduction in BB performance of 0.1 points in standard deviation (SD), the risk of falling increased by up to 0.05%. According to our analysis, members of the non-faller group had better postural control than those of the faller group. The difference between the groups was considerably large (Δ = 2.39 points). However, considering the ideal cutoff score of 25 out of 40 on the FAB scale, fallers and non-fallers indicated preserved postural control [40]. It is worth mentioning that falls are multifactorial events, determined by the joint action of extrinsic and intrinsic factors [47]. 

With aging, gait patterns (e.g., speed, cadence, stride width), coordination, muscle strength, respiratory endurance, and balance tend to deteriorate [48]. In the case of GS and BB, this process is caused by several factors, such as changes in cognitive functions [49,50], somatosensory damage [51], vestibular problems [52], as well as neuromuscular problems responsible for muscle weakness [53]. Thus, compared to young individuals, older adults move more slowly, which makes them more vulnerable to falls, especially in challenging situations, such as those that require the simultaneous processing of two or more motor and cognitive tasks [54,55]. Older adults also have difficulty performing postural corrections accurately and quickly in situations of imbalance caused by external disturbances [56,57]. In this perspective, our findings were in line with the literature suggesting the promotion of PA as a valuable strategy to increase the performance of GS and BB (physical functions), benefiting the reduction of the risk of falls [58].

In a meta-analysis study, Chase et al. [59] showed that PA might be able to improve the physical functions of older adults. Furthermore, it is known that there is a bidirectional relationship between PA and physical function [60]. Therefore, the improvement in physical functions reflects positively on PA levels, which comes from the perception of improvement in motor skills (i.e., GS and BB). This makes the individuals more motivated and willing for everyday activities and encourages them to remain physically active [9]. As a result, the individual perceives an improvement in the general state of his health, which consequently benefits his HRQoL [61]. In our analysis, the SIE1 and SIE4 pathways showed the benefits of increasing GS and BB (physical functions) in the link between PA and HRQoL in dependence on falls (see Figure 1). We found that an increase in the standard deviation (SD) of 0.1 m/s in the GS performance represented an increase in HRQoL of up to 13.75%. At the same time, an increase in the standard deviation (SD) of 0.1 m/s in the execution of the BB, meant an increase in HRQoL of up to 0.65%.

Studies have reported that a high GS is able to increase the perception of HRQoL in the older adult population [35,62], helping to maintain levels of independence [63]. An improvement in GS can promote beliefs about one’s own abilities to deal with the environment in which one lives, which is important for strengthening self-confidence and generating new perspectives on life during aging [64]. Experimental studies have also shown the effects of BB training on HRQoL in community-dwelling older adults [65,66]. Regarding the relationship between PA and HRQoL, with BB as one of the mediators, other factors must be considered as mediators or moderators. A current study pointed out that in the relationship between PA and HRQoL, BB and lower limb strength mediated the relationship by approximately 47% and 39.6%, respectively [30]. Moreover, unipedal support and lower limb strength were identified in older adults (75 years) as mediators in the relationship between walking habits and HRQoL [67]. The outcomes attest that one or two variables cannot explain falls, in addition to the fact that the association between PA and HRQoL is widely considered. Another point to highlight in the relationships among PA, GS, BB, and falls is the association between HRQoL and fear of falling (FOF). FOF is considered a psychological risk factor for intrinsic falls, which is combined with the others (i.e., GS and BB) related to falls [68]. It is known that after a fall, the older adult develops the fear of falling again [69], becoming more cautious about walking and performing ADLs. Thus, due to the intentional reduction of the GS or BB, the ability of the older adult person to execute movements quickly due to external mechanisms (i.e., collisions with objects and people, tripping and slipping) is limited, increasing their chances of falling [70]. A systematic review study showed a negative relationship between FOF and HRQoL of individuals aged ≥60 years [71], particularly the discredit in physical beliefs. In our study, 44.4% of fallers were recurrent. Thus, some participants may have developed FOF, and this feeling has reduced GS [72] and BB control [73], co-participating in the occurrence of new fall events, in turn reducing HRQoL [67]. Our findings also corroborate a previous investigation that pointed to GS, unipodal support, and lower limb strength in older adults (75 years old) as mediators in the relationship between walking habits and HRQoL [67], as well as an increase in HRQoL due to increased PA levels in the bilateral relationship with an improvement in physical functions, which includes GS and BB [30,34].

Several limitations of the current study are important to note. First, the use of a cross-sectional design did not allow us to attribute causalities in mediation analyses. Thus, it is suggested that future investigations adopt longitudinal designs or experimental studies to generalize the results presented here and, in turn, expand our understanding of these study variables during the aging process. Second, it should be considered that the main characteristic of the sample (older adult inhabitants of Madeira Island) may have influenced the general findings. Thus, we suggest caution in generalizing the results. Third, it should be considered that the participants in this study were recruited from different locations, including sports clubs and associations: It is possible that the PA levels of these older adults were higher than those of the other participants. Fourth, it should be considered that, although the Baeck questionnaire is used worldwide to measure PA levels in older adults, there are instruments with better validity and reliability scores, such as the Physical Activity Scale for the Elderly (PASE) [74]. This fact does not detract from our results; however, it serves as a suggestion for future investigations. Our study has a strong point. It is the first to investigate the mediating role of GS, BB, and falls in the association between PA and HRQoL, bringing to light information about the proportion of this mediation. Moreover, we encourage the development of new investigations that evaluate the association between PA and HRQoL, mediated by GS, BB, and falls, considering potential moderators such as fear of falling, sex, and age to determine differences.

## 5. Conclusions

Our results suggested that the association between PA and HRQoL in vulnerable older adults was partially mediated by GS, BB, and the frequency of fall events. Based on parallel serial analysis mediation procedures, we detailed in proportional terms the role that each of the three mediating variables plays in the association between PA and HRQoL, in addition to the proportion of the total effect of PA on HRQoL. Moreover, the parallel–serial mediation pathway model indicated that total PA played a crucial role in improving physical functions (i.e., GS and BB), counteracting the negative effects related to age, capable of increasing the risk of falling, benefiting, in turn, the perception of HRQoL. From this perspective, when it comes to falls in the older adult population, the results can provide useful information to professionals in the clinical area, helping them to identify and monitor levels of PA and HRQoL.

## Figures and Tables

**Figure 1 ijerph-19-14135-f001:**
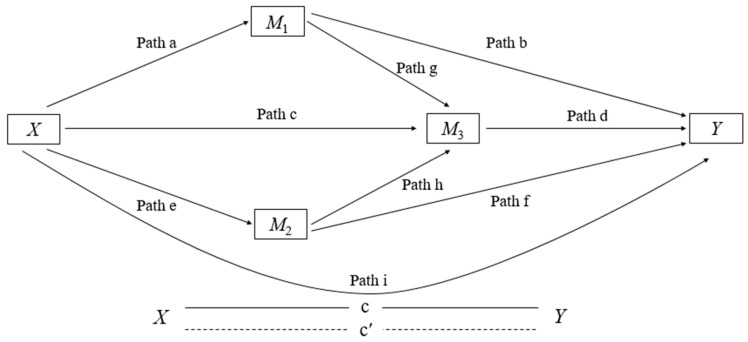
Parallel model of serial mediation to determine the mediating effect of GS, BB and falls in the association between PA and HRQoL. Path (a) = association between independent variable PA (*X*) with GS mediator (*M*_1_), Path (b) = association between GS mediator (*M*_1_) with dependent variable HRQoL (*Y*), Path (e) = association between independent variable PA (*X*) with BB mediator (*M*_2_), Path (f) = association between BB mediator (*M*_2_) with dependent variable HRQoL (*Y*), Path (g) = association between GS mediator (*M*_1_) with mediator Falls (*M*_3_), Path (h) = association between BB mediator (*M*_2_) with mediator Falls (*M*_3_), Path (c) = association between independent variable PA (*X*) with Falls mediator (*M*_3_), Path (d) = association between Falls mediator (*M*_3_) with dependent variable HRQoL (*Y*), and Path (i) represents c’ = direct effect (*X* − *Y*).

**Figure 2 ijerph-19-14135-f002:**
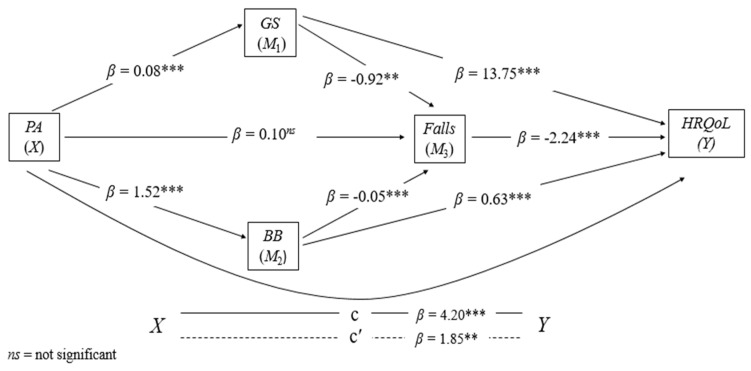
Results of the parallel–serial model of mediation of GS, BB and falls in the relationship between PA and HRQoL. The mediation hypothesis was tested by the bootstrap method with bias correction. Number of bootstrap samples = 5000. samples to calculate confidence intervals (95%), when the CI did not include 0 (zero). The products of the simultaneous regressions were presented by values of betas (β). *X* = PA: independent variable, *Y* = HRQoL: dependent variable.; ** *p* < 0.01; *** *p* < 0.001. Note: PA: physical activity; GS: gait speed; BB: body balance; HRQoL: health-related quality of life.

**Table 1 ijerph-19-14135-t001:** Main characteristics of the sample.

Variable	Full Sample (*n* = 619)	Faller (*n* = 225)	Non-Faller (*n* = 394)	*p*-Value
Age (years)	69.5 ± 5.6	70.05 ± 5.92	69.18 ± 5.44	0.066
60–69	294 (47.5)	104 (84.4)	190 (48.2)	
70–79	303 (48.9)	110 (48.8)	193 (449.0)	
80–89	22 (3.5)	11 (4.8)	11 (2.8)	
Sex				<0.001
Woman	314 (50.7)	153 (68.0)	161(40.9)	
Men	305 (50.3)	72 (32.0)	233 (59.1)	
Falls				<0.001
1	114 (18.4)	114 (50.6)	-----	
2–3	55 (8.9)	55 (24.4)	-----	
4–6	25 (4.4)	25 (11.1)	-----	
7–10	20 (3.2)	20 (8.9)	-----	
Medication	4.66 ± 0.03	4.68 ± 0.5	4.61 ± 0.6	0.395
Comorbidities				
Vision	486 (78.5)	204 (90.6)	282 (71.6)	<0.001
Hearing	196 (31.6)	76 (33.7)	120 (30.4)	0.228
Hypertension	408 (65.9)	154 (68.4)	254 (64.6)	0.022
Diabetes	224 (36.2)	104 (46.2)	120 (30.4)	0.122
Musculoarticular		23 (10.2)	23 (5.8)	0.035
Anthropometry				
Height (cm)	159.05 ± 8.69	156.43 ± 8.37	160.79 ± 8.26	<0.001
Weight (kg)	74.77 ± 13.06	73.49 ± 12.79	75.88 ± 13.00	0.027
BMI (kg/m^2^)	29.51 ± 4.34	29.99 ± 4.52	29.31 ± 4.29	0.065
GS	1.24 ± 0.25	1.20 ± 0.25	1.28 ± 0.24	<0.001
BB	30.53 ± 0.26	29.40 ± 7.62	31.79 ± 6.72	<0.001
HRQoL	68.57 ± 17.96	62.72 ± 18.78	71.88 ± 17.22	<0.001
Total PA (score)	7.30 ± 1.23	7.29 ± 1.14	7.35 ± 1.26	0.572

Values are mean ± SD or n and percentages in brackets; kg: kilogram; cm: centimeter; BMI: body mass index; kg/m^2^; GS: gait speed; BB: body balance; HRQoL: health-related quality of life; PA: physical activity. *p* < 0.01; *p* < 0.001.

## Data Availability

The data presented in this study are available upon request from the corresponding author.

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
