# Peer review of "Exploring Mediation Effects of Gait Speed, Body Balance, and Falls in the Relationship between Physical Activity and Health-Related Quality of Life in Vulnerable Older Adults"

_ijerph, 2022, doi:10.3390/ijerph192114135_

Round 1

Reviewer 1 Report

The reviewer would like to thank the authors for the work done in the paper and only has minor comments and questions. I do not have any specific comments and only general ones. 

- The reviewer would like to know why the authors have chosen the different questionnaires mentioned in the paper. 

For example, the Baecke questionnaire has only adequate correlation (r=0,54) for PA in elderly (hertogh et al. 2018) and even thought the construct validity is excellent, the convergent validity is quite poor. The PASE (Physical activity Scale for Elderly) has been presented with better validity and reliability scores (Washburn et al. ) This aspect should be mentioned in the limitation part of the paper. 

- What were the criteria defining fallers vs non fallers? In the table1, one fall was already considered in the fallers group. There is a battery of test usually used for this (fall risk assessment tool). This aspect should be describe in the Methodology part.

- Linked to the high statistical approach, it would be great to have validity and reliability of the test mentioned. 

Author Response

Dear Reviewer, we are grateful for all the comments, and are available for future clarifications and/or corrections.

* Changes referring to the last review of the manuscript were carried out in the text using Microsoft Word's built-in track changes function.

  1. The reviewer would like to know why the authors have chosen the different questionnaires mentioned in the paper. For example, the Baecke questionnaire has only adequate correlation (r=0,54) for PA in elderly (Hertogh et al. 2018) and even thought the construct validity is excellent, the convergent validity is quite poor. The PASE (Physical activity Scale for Elderly) has been presented with better validity and reliability scores (Washburn et al. ) This aspect should be mentioned in the limitation part of the paper

Reply

Dear Reviewer, Thank you for your excellent suggestion. We added a third point in the limitations section on the PASE questionnaire (lines 421-425).

  1. What were the criteria defining fallers vs non fallers? In the table1, one fall was already considered in the fallers group. There is a battery of test usually used for this (fall risk assessment tool). This aspect should be describe in the Methodology part.

Reply

Dear Reviewer, this information was better highlighted in its own section (2.2.1), in lines 129-134.

  1. Linked to the high statistical approach, it would be great to have validity and reliability of the test mentioned.

Reply

Dear Reviewer, as requested, detailed information about the instruments has been included in their respective sections.

Reviewer 2 Report

In the background, the authors should discuss in depth the findings of the previously published article on this project (https://www.mdpi.com/2077-0383/11/16/4857). It is necessary to justify the originality of the current manuscript.

Author Response

Dear Reviewer, we are grateful for all the comments, and are available for future clarifications and/or corrections.

* Changes referring to the last review of the manuscript were carried out in the text using Microsoft Word's built-in track changes function.

  1. In the background, the authors should discuss in depth the findings of the previously published article on this project (https://www.mdpi.com/2077-0383/11/16/4857). It is necessary to justify the originality of the current manuscript.

Reply

Dear Reviewer, thank you for your observation. The required information has been added to the end of the Introduction section (lines 89-94). We are available for future adjustments, if necessary.

Round 2

Reviewer 2 Report

Thank you for considering my remarks